# Two Types of Support for Redistribution of Wealth: Consistent and Inconsistent Policy Preferences

**Naoki Sudo** 

Department of Political Studies, Faculty of Law, Gakushuin University, Tokyo 171-8588, Japan;
naoki.sudo@gakushuin.ac.jp

**Abstract:** This article aims to clarify the latent structure of support for redistribution. To this end, the author analyzed data from the *National Survey of Social Stratification and Social Mobility in 2015* (SSM 2015), which was conducted in Japan, using finite mixtures of regression models. The results revealed that the population could be categorized into two latent groups: one that determines preferences for social policies based on self-interest and another that does so based on ideology. Surprisingly, the results also showed that, compared to those who supported redistribution of wealth based on ideology, those who supported them based on self-interest were more likely to hold inconsistent preferences (e.g., simultaneous support for redistribution of wealth and free-market competition). This observation implies that, even when individuals want to determine their policy preferences rationally, they often do not have enough information to correctly assess the influence of each social policy on their self-interest.

**Keywords:** social policy; redistribution; preferences; self-interest; ideology; Japan

## 1. Introduction

As Koos and Sachweh [1] pointed out, support for redistribution is not necessarily incompatible with support for free-market competition. Rather, support for redistribution might serve as compensation for "support for free competition as a dominant ideology." However, if individuals rationally decide their preferences for social policies based on self-interest, simultaneous support for redistribution of wealth and free-market competition appears contradictory and irrational.

Nevertheless, many previous studies have clarified that individuals tend to form their preferences for social policies based on self-interest. For example, it is well known that low income and unstable employment have positive influences on support for redistribution [2–11], as redistributive policies can expected to yield various benefits for such individuals. Moreover, it is also known that, when immigrants are viewed as competitors for the benefits of such policies, an increase in the number of immigrants has a negative influence on support for them [12–15]. These observations suggest that individuals tend to determine their preferences for social policies based on self-interest. Therefore, it can be said that support for redistribution of wealth (and free-market competition) poses a problem for social researchers.

On the other hand, other studies have revealed that preferences regarding social policies are influenced by not only self-interest, but also other factors, such as social norms [16–19], group interests [20], social image [21–25], and social regimes [26–32]. As a result, while an increase in the number of immigrants tends to have a negative effect on support for redistribution of wealth, it might have a positive (or, at least, neutral) effect on support for such policies in different social contexts [33–36]. Similarly, while some studies have shown that social inequality has a positive effect on support for redistribution [37–41], others have shown that social inequality might not have such an effect on support for such policies in some cases [42–46]. Thus, it has been clarified that support for redistribution of wealth has more complicated characteristics than might be expected. In analyzing the



(latent) structure of support for redistribution of wealth, support for them should not be interpreted based on only a single social mechanism, such as that of self-interest. Rather, it should be assumed that there are various mechanisms that generate support for redistribution of wealth within a society.

Therefore, the author of this study aims to clarify the latent structure of support for the redistribution of wealth and, in doing so, specify the process of forming preferences for social policies. Hence, we clearly understand why inconsistent preferences for social policies can be generated by rational individuals, and why their preferences tend not to influence actual social policies. Moreover, on the basis of this finding, we will find a way to overcome problems deduced from seemingly irrational preferences for social policies.

## 2. Theory and Hypotheses

To explain the complexity of support for redistribution of wealth, the author of this study assumed that various groups with various mechanisms for determining preferences regarding social policies coexisted within societies. Even in countries that have implemented redistributive policies, there are class-based cleavages in attitudes toward them [47]. This suggests that individuals within a society do not obey a single or universal mechanism for determining preferences related to social policies. Therefore, the assumption that there are various mechanisms for determining preferences related to social policies within a society is more plausible.

Specifically, it is assumed that a population can be categorized into two groups: one that forms its social policy preferences based on self-interest and another that does so based on ideology, social norms, or the opinions of authorities.

Members of the former group were those who determined support for redistribution of wealth based on self-interest. Therefore, if they thought that redistributive policies might yield significant benefits for them, they could be expected to view such policies positively. However, if they thought that redistributive policies would impose heavy burdens that would outweigh their benefits, they could be expected to support market-oriented policies instead. As a result, it was predicted that their preferences for social policies would be consistent from the standpoint of self-interest. Additionally, the supporters of redistribution of wealth among the former group were more likely to be socially disadvantaged people who could be expected to receive various benefits from such policies.

On the other hand, the members of the latter group were those who determined support for redistribution of wealth based on ideology, social norms, or the opinions of authorities (e.g., political leaders, religious leaders, intellectuals, and so on). For example, if authorities they trusted espoused redistributive policies, they could be expected to support such policies too, regardless of their impact on their interests. Conversely, if such authorities espoused market-oriented policies, they could also be expected to support such policies, regardless of their impact on their interests. In this case, it could be expected that their preferences for social policies might lack consistency from the point of view of self-interest. Additionally, the supporters of redistribution of wealth among the latter group could be expected to include socially advantaged, as well as socially disadvantaged, people.

Based on these inferences, some hypotheses regarding the process of forming preferences regarding social policies were constructed. First, regarding the coexistence of various social mechanisms determining social policy preferences, the following hypothesis can be formalized.

**Hypothesis (H1).** *The population can be categorized into two groups. Individuals in one group tend to hold preferences for social policies that correspond to their interests, as they tend to examine social policies by considering their personal benefits and burdens. On the other hand, individuals in the other group might have a preference for social policies that do not correspond to their interests, as they tend to determine for support such policies based on ideology, social norms, and the opinions of authorities.*

It must be noted that Hypothesis 1 does not refer to differences in preferences themselves among people. Rather, it focuses on differences in the psychological mechanism that generates such preferences.

We can observe that there are individuals who share social environments (demographic characteristics, socio-economic status, and geographic residence). If they obey the same mechanism for generating preferences regarding social policies, it can be expected that they will have preferences for similar social policies. However, if they obey varied mechanisms, it can be expected that they may have different preferences regarding them. Thus, differences in preferences regarding social policies among them should be interpreted as being caused by social mechanisms and should not be deduced from social environments alone.

Additionally, the following hypothesis related to group decision-making preferences regarding social policies based on self-interest can be formalized.

**Hypothesis (H2).** *Compared to supporters of redistribution of wealth that belong to the group that determines its preferences regarding social policies based on ideology, supporters of redistribution of wealth that belong to the group that determines its preferences regarding social policies based on self-interest are less likely to support free-market competition.*

As mentioned above, individuals who belong to the group that determines its preferences regarding social policies based on self-interest tend to have consistent preferences from the standpoint of self-interest. Therefore, if they support redistributive policies based on self-interest, it can be predicted that if they do not support free-market competition, it is because it is not in their best interest to do so.

Nevertheless, it is already well-known that many people support market-oriented policies while also supporting redistributive policies. This implies that many people have contradictory preferences regarding social policies. In other words, while some supporters of redistribution of wealth (who belong to the group that determines its preferences regarding social policies based on self-interest) do not support free-market competition, other supporters of redistribution of wealth (who belong to the group that determines its preferences regarding social policies based on ideology) do. Thus, by assuming the presence of authoritarian attitudes and discrepancies among authorities' opinions, the contradictory preferences held by supporters of redistribution of wealth based on authorities' opinions can be explained as follows. Even among political and religious leaders and intellectuals, there are various discrepancies in opinions regarding social policies. In a world that has become highly complicated as a result of globalization, such discrepancies can be found easily. Without carefully considering them, individuals who rely on the opinions of authorities to guide their own might unintentionally form contradictory beliefs regarding social policies. For example, some who belong to the group that forms its preferences regarding social policies based on authorities' opinions may support redistributive policies based on a religious leader's opinion while supporting market-oriented policies based on that of a political leader. As such individuals often fail to adequately grasp the impact of each social policy, they may not feel uneasy about holding contradictory opinions.

From theses inferences, the following hypothesis related to group decision-making preferences regarding social policies based on ideology can be formalized.

**Hypothesis (H3).** *Compared to supporters of redistribution of wealth who belong to the group that determines its preferences regarding social policies based on self-interest, supporters of redistribution of wealth who belong to the group that determines preferences regarding such policies based on ideology are more likely to support free-market competition.*

As previous studies have shown, the dynamics of support for redistribution of wealth are highly complex. However, such complicated dynamics can be explained by assuming the coexistence of various psychological mechanisms that generate preferences regarding social policies within a society. The rational aspect of support for redistribution of wealth is a consequence of a psychological mechanism that generates policy preferences based on self-interest, while the contradictory aspects of support for such policies are consequences of a psychological mechanism that generates policy preferences based on ideology.

## 3. Data and Methods

### 3.1. Data

In order to examine these hypotheses related to individual preferences regarding social policies, this study was based on data from a representative national survey, the National Survey of Social Stratification and Social Mobility in 2015 (SSM 2015), which was conducted in Japan. The SSM 2015 is one of several SSM surveys conducted by Japanese sociologists every decade since 1955 with the objective of understanding the trends and changes in social inequality in Japanese society [48]. The survey is also conducted with the aim of exploring social stratification and social mobility in Japan and contains rich information related to the social attitudes of respondents.

The SSM 2015 was conducted from January to August 2015. The population of respondents was comprised of Japanese citizens between the ages of 20–80 years. The sample was selected from the *Jyumin-kihon-daicho* (residence registers administered by each municipality) based on a multi-stratified random sampling method. The survey method included a combination of personal interviews and replacement methods. The information about the respondents' demographic characteristics and socio-economic status was collected through personal interviews, and the information related to their social attitudes was obtained using a questionnaire.

The total number of respondents to the SSM 2015 was 7817, and the response rate was 50.1%. As some cases had missing values for target variables, 7021 cases were used for analysis in this study. Seemingly, this response rate was not very high. However, the response rates of academic surveys conducted in Japan have been declining over the last few decades [49], often falling below 50% in recent years. Therefore, the response rate to the SSM 2015 is not particularly low when compared to those of other academic surveys conducted in Japan. Rather, as the data from the SSM 2015 include rich information on the demographic characteristics and socio-economic status of the respondents, Japanese sociologists regard it as one of the most reliable datasets in Japan.

On the other hand, the data from the SSM 2015 suffer from a few limitations regarding the attempt to understand the latent structure of support for redistribution of wealth in general. As the survey includes only Japanese living in Japan in 2015, the SSM 2015 provides cross-sectional data for only a single country. It is, therefore, difficult for researchers to deduce generalized findings regarding differences between societies. Moreover, it is not possible to examine historical changes in the latent structure of support for redistribution of wealth only by using the SSM 2015.

### 3.2. Variables

#### 3.2.1. Dependent Variables

The dependent variable in the analysis in this study is "support for redistribution." In the SSM 2015, all respondents were asked about support for redistribution of wealth in the following statement:

*The following opinions concern how society should work. What do you think about each statement?*

*Rather than protecting free-market competition, it is more important to eliminate differences among people.*

Since the respondents were asked about this item just after they were asked about the item related to "support for free-market competition," it is naturally assumed that, even though this item does not refer to "wealth," the respondents perceived it as centering on differences in wealth among people. The respondents were requested to choose one of five alternatives (Agree, Somewhat agree, No opinion either way, Somewhat disagree, Disagree). In this analysis, the variable "support for redistribution" was treated as a continuous variable (Agree = 5, . . . , Disagree = 1). Similarly, respondents were asked about support for free-market competition in the following sentence:

*To the extent that opportunities are equally available, we must accept the disparity in wealth that results from competition.*

The variable of support for free-market competition also was treated as a 5-point-scale variable.

### 3.2.2. Independent Variables

The key independent variables in this analysis were those related to ideology and authoritarian attitudes because it was assumed that the inclination to favor authoritarian attitudes would generate a subpopulation that would support redistribution of wealth regardless of self-interest. In this analysis, an authoritarian personality was measured by the following two items: "It generally works out best to keep on doing things the way they have been done before" and "In this complicated world, the only way to know what to do is to rely on leaders and experts." Then, the respondents who agreed with these items (respect for old ways and reliance on leaders/experts) were classified as individuals who held ideology and authoritarian attitudes. This was also treated as a 5-point-scale variable (Agree = 5, ... , Disagree = 1).

### 3.2.3. Control Variables

The influence of the respondents' demographic characteristics and socio-economic status needed to be controlled. To control the influence of these demographic characteristics, the variables of age, gender, and marital status were used. Age was treated as a continuous variable (from 20 to 80) and gender was treated as a dummy variable ("man" coded as "0" and "woman" coded as "1"). Marital status was divided into three categories (married, unmarried, and divorced/bereaved), and dummy variables were made for each category.

Similarly, to control the influence of socio-economic status, the variables of educational level, employment status, occupation, and household income were employed. Educational level was divided into three categories (primary, middle, and higher education), and dummy variables were made for each group. Employment status (self-employed, regular employed, non-regular employed, job seeker, and no job) and occupation type (upper white collar, lower white collar, and blue collar) were regarded as categorical variables, and household income was regarded as a continuous variable. As the distribution of household income tended to be skewed, the household income variable was used after log-transformation.

### 3.3. Analytic Strategy

In this study, the data from the SSM 2015 were analyzed using finite mixtures of regression models [50–52]. Using these models, the subpopulations (latent classes) could be extracted from the entire population, and the mechanisms that generated support for redistribution of wealth could be specified for each subpopulation. Here, a finite mixture density of regression models with K latent classes is represented by the following equation [50]:

$$h(y|x,\theta) = \sum_{k=1}^{K} \pi_k f_k(y|x,\theta_k), \qquad (1)$$

where $\Theta$ is the vector of all parameters for the mixture density $h$. The dependent variable is $y$, and the independent variables are $x$. The latent class-specific density function is $f_k$. The latent class-specific parameters are given by $\theta_k = (\beta'_k, \phi_k)$, where $\beta_k$ is the regression coefficient and $\phi_k$ is the dispersion parameter. For the latent class weights $\pi_k$, it holds that

$$\sum_{k=1}^{K} \pi_k = 1 \wedge \pi_k > 0, \vee k. \qquad (2)$$

When the data from the SSM 2015 were analyzed using finite mixtures of regression models, the parameters and coefficients were estimated using the flexmix package in Software R [50,51,53].

After specifying the subpopulations to which the various regression models predicting support for redistribution of wealth applied, the author compared the differences in members' characteristics for each subpopulation, with a particular focus on support for free-market competition, ideological attitudes (respect for old ways, reliance on leaders/experts), demographic characteristics, and socio-economic

status. According to the hypotheses of this study, the members of one of the subpopulations could be expected to be more likely to hold ideological attitudes and support redistribution of wealth as well as free-market competition, while members of the other subpopulation could be expected to be less likely to hold ideological attitudes and more likely to be consistent in their support for social policies. Moreover, it was thought that if the theoretical explanation in this study of supporters of redistribution of wealth were correct, the differences in socio-economic status between the supporters of such policies among the ideological group and those among the non-ideological group could be identified. In other words, it was hypothesized that people who hold contradictory opinions regarding social policies are less likely to be socially advantaged, while those who hold consistent opinions regarding them are more likely to be socially advantaged. Thus, the validity of the theoretical explanation offered in this study was examined by comparing the socio-economic status of the two subpopulations.

## 4. Results

### 4.1. Descriptive Statistics

First, the basic characteristics of the cases used in the analysis were examined. Table 1 presents the results of the descriptive statistics for all variables used in this analysis. For continuous variables (age, household income, support for redistribution of wealth, support for free-market competition, respect for old ways, and reliance on leaders/experts), the values of the arithmetic mean, standard deviation, minimum value, and maximum value are shown. For dummy variables, on the other hand, only the ratios of each variable in the sample are shown. Moreover, the number of respondents in this analysis was 7201. It should be noted that the variable of household income (log) had many missing values among the respondents. If respondents with missing values for household income (log) had been excluded from the analysis of this study, the number of the effective respondents would have been reduced to 5104. To avoid this reduction, the missing values for household income (log) were substituted with predicted values, which were calculated based on the Markov chain Monte Carlo method using RStan [54].

**Table 1.** Descriptive statistics (overall).

| Variables | Mean/Ratio | SD | Min | Max |
|---|---|---|---|---|
| Support for redistribution | 3.30 | 1.06 | 1 | 5 |
| Support for free competition | 3.43 | 1.12 | 1 | 5 |
| Respect for old ways | 2.19 | 1.02 | 1 | 5 |
| Reliance on experts/leaders | 2.57 | 1.10 | 1 | 5 |
| Age | 52.48 | 16.12 | 20 | 80 |
| Women | 0.53 | | | |
| Primary education | 0.12 | | | |
| Middle education | 0.62 | | | |
| Higher education | 0.26 | | | |
| Married | 0.72 | | | |
| Unmarried | 0.17 | | | |
| Divorced/bereaved | 0.11 | | | |
| Self employed | 0.15 | | | |
| Regular employee | 0.32 | | | |
| Non-regular employee | 0.20 | | | |
| No job | 0.31 | | | |
| Job seeker | 0.03 | | | |
| Upper white collar | 0.17 | | | |
| Lower white collar | 0.23 | | | |
| Blue collar | 0.27 | | | |
| Household income (log) | 6.12 | 0.70 | 0.00 | 8.94 |
| N = 7021 | | | | |

According to the figures presented in Table 1, the mean value of support for redistribution of wealth is 3.30. This value is greater than 3.0, implying that the Japanese tend to support it. However, the figures in Table 1 also indicate that the mean value of support for free-market competition is greater than 3.0. This means that the Japanese tend to support free-market competition as well. Furthermore, the mean value of support for free-market competition surpasses that of redistribution of wealth. When examining the latent structure of support for the latter, it is important to consider the relationship between support for redistribution of wealth and support for free-market competition.

In this analysis, the key dependent variables were those related to ideological attitudes. According to the data presented in Table 1, the mean of the variable "respect for old ways" is 2.19 and that of "reliance on leaders/experts" is 2.57. As both values are under 3.0, the results imply that the Japanese tend to disagree with ideological attitudes. Therefore, it can be concluded that people with ideological attitudes are not a majority in Japanese society.

## 4.2. The Results of the Finite Mixture of Regression Models

Table 2 presents the results of the finite mixtures of regression models predicting support for redistribution of wealth based on the SSM 2015 (the model without the income variable was also examined. See Appendix A). Model 1 in Table 2 is a model with one latent class, which substantially resembles a simple multiple regression model. On the other hand, Model 2 in Table 2 is a model with two latent classes: Class 1 and Class 2. Class 1 in Model 2 shows a statistically significant negative effect of support for free-market competition and a statistically significant positive effect of respect for old ways on support for redistribution of wealth. Conversely, Class 2 in Model 2 shows a statistically significant positive effect of support for free-market competition and a statistically significant positive effect of reliance on leaders/experts on support for redistribution of wealth. Clearly, the effects of key independent variables are quite different in Classes 1 and 2. Then, it needs to be noted that models in Table 2 include "support for free-market competition" as an independent variable. If it is assumed that these models predict support for social policies, this means that the variance in endorsements of social policies is partially explained by their support. Considering the possibility of endogeneity bias, therefore, the highly significant effects of support for free-market competition on support for the redistribution of wealth should not be overestimated Nevertheless, discrepancies in the directions of the effects between Classes 1 and 2 should be paid more attention. The direction of the effect of support for free-market competition on support for the redistribution of wealth in Class 1 is opposite to that of the effect in Class 2. This implies that Classes 1 and 2 have different internal structures for support for social policies.

Table 2 also shows the values of information criteria for each latent class. The value of the Akaike Information Criterion (AIC) in Model 2 is less than that of the AIC in Model 1. Moreover, the value of the Bayesian Information Criterion (BIC) in Model 2 is also lower than that of the BIC in Model 1. Based on the AIC and BIC, it can be concluded that Model 2 is better fitted to the data from the SSM 2015 than Model 1. Furthermore, as mentioned above, there are distinct differences in the effects of key independent variables (support for free-market competition, respect for old ways, and reliance on leaders/experts) between Class 1 and Class 2 in Model 2. This finding partially supports Hypothesis 1 of this study; therefore, it can be assumed that various mechanisms generating support for redistribution of wealth coexist within a society.

Between Classes 1 and 2 in Model 2, there are also differences in the effect of socio-economic status on preferences regarding social policies. Regarding educational level, Class 2 shows a statistically significant negative effect of high educational attainment on support for redistribution of wealth, which corresponds to the self-interest hypothesis. Similarly, regarding occupation, Class 2 shows a statistically significant positive effect of less-secure employment status on support for such policies, which also corresponds to the self-interest hypothesis. Finally, the negative effect of household income on support for redistribution of wealth in Class 2 is larger than in Class 1. Taken together, it can be

observed that Class 2 reflects the mechanism that generates support for redistribution of wealth based on self-interest.

On the other hand, the effect of socio-economic status on support for redistribution of wealth in Class 1 is relatively weak. The effects of having only a primary education or being self-employed in Class 1 certainly have statistical significance. However, the effect of having a primary education is negative, which is the opposite of what would be expected based on the self-interest hypothesis. Additionally, the rank of the self-employed within the employment status hierarchy is unclear compared to that of regular employees, non-regular employees, and jobseekers. While socio-economic status in Class 1 has a weak effect on support for redistribution of wealth, the variable of respect for old ways in Class 1 has a statistically significant effect on it. These observations suggest that Class 1 reflects the mechanism that generates support for redistribution of wealth based on ideology. Thus, Hypothesis 1 of this study was supported by the analytical results based on the SSM 2015.

**Table 2.** Results for the finite mixtures of regression models predicting support for redistribution.

| | Model 1 | | | Model 2 | | | | |
| | | | | Class 1 | | | Class 2 | |
| Variable | Coeff. | | SE | Coeff. | | SE | Coeff. | | SE |
|---|---|---|---|---|---|---|---|---|---|
| (Intercept) | 4.220 | *** | 0.156 | 5.384 | *** | 0.196 | 2.566 | *** | 0.359 |
| Age | 0.004 | *** | 0.001 | 0.002 | | 0.001 | 0.004 | | 0.002 |
| Women | 0.134 | *** | 0.027 | 0.136 | *** | 0.034 | 0.102 | | 0.057 |
| (Married) | | | | | | | | | |
| Unmarried | 0.018 | | 0.038 | −0.005 | | 0.046 | 0.062 | | 0.082 |
| Divorced/bereaved | 0.046 | | 0.042 | 0.030 | | 0.056 | 0.061 | | 0.087 |
| Primary education | 0.149 | *** | 0.041 | 0.121 | * | 0.056 | 0.048 | | 0.081 |
| (Middle education) | | | | | | | | | |
| Higher education | −0.131 | *** | 0.030 | −0.037 | | 0.037 | −0.278 | *** | 0.067 |
| (Regular employee) | | | | | | | | | |
| Self employed | −0.082 | * | 0.040 | −0.133 | ** | 0.050 | 0.019 | | 0.087 |
| Non-regular employee | −0.003 | | 0.037 | −0.020 | | 0.046 | −0.005 | | 0.079 |
| No job | 0.015 | | 0.043 | −0.053 | | 0.054 | 0.130 | | 0.093 |
| Job seeker | 0.102 | | 0.078 | −0.040 | | 0.089 | 0.291 | | 0.174 |
| Upper white | −0.045 | | 0.039 | −0.080 | | 0.049 | −0.033 | | 0.085 |
| (lower white) | | | | | | | | | |
| Blue collar | 0.153 | *** | 0.035 | 0.046 | | 0.044 | 0.241 | ** | 0.078 |
| Household income [log] | −0.092 | *** | 0.021 | −0.056 | * | 0.025 | −0.144 | ** | 0.046 |
| Support for free competition | −0.259 | *** | 0.011 | −0.649 | *** | 0.024 | 0.373 | *** | 0.051 |
| Respect for old ways | 0.058 | *** | 0.013 | 0.068 | *** | 0.017 | 0.013 | | 0.029 |
| Reliance on experts/leaders | 0.038 | ** | 0.012 | 0.020 | | 0.015 | 0.064 | * | 0.026 |
| Π | | | | | 0.620 | | | 0.380 | |
| AIC | | 19,773.99 | | | 19,119.17 | | | | |
| BIC | | 19,897.41 | | | 19,372.86 | | | | |
| N = 7021 | | | | | | | | | |

\* $p < 0.05$, \*\* $p < 0.01$, \*\*\* $p < 0.001$.

Interestingly, the effect of support for free-market competition on support for redistribution of wealth in Class 2 is statistically significant and positive. In other words, supporters of redistribution of wealth based on self-interest are more likely to hold inconsistent beliefs regarding social policies, and they tend to support redistributive policies and market-oriented ones simultaneously. This indicates that, while they support redistributive policies based on self-interest, their judgements regarding what is in their best interest might be inconsistent. It must be noted here that the effect of reliance on leaders/experts on support for redistribution of wealth is also statistically significant and positive in Class 2. Thus, if the opinions of leaders and experts are fragmented and inconsistent with each other because a society is highly developed and complex, it can be predicted that individuals' judgements regarding their best interests will become obscured and inconsistent. As a result, it can be observed that

the members of Class 2 tend to have inconsistent beliefs regarding social policies. This finding does not correspond to Hypothesis 2 of this study; however, it appears to have thought-provoking implications.

Compared to Class 2, the effect of support for free-market competition on support for redistribution of wealth in Class 1 is statistically significant but negative. In other words, supporters of such policies based on ideology are more likely to hold consistent beliefs regarding social policies. For example, if they do not support market-oriented policies, they tend to support redistributive policies. This indicates that they determine their preferences for social policies based on a clear principle, and their judgements regarding social policies tend to be consistent regardless of their personal interests or leaders'/experts' opinions. While the effect of respect for old ways on support for redistribution of wealth in Class 1 is statistically significant, that of reliance on leaders/experts is not.

### 4.3. Basic Characteristics of Supporters of Redistribution

Table 3 shows the descriptive statistics of supporters of redistribution of wealth in Classes 1 and 2. Supporters of redistribution of wealth were defined as those who chose "Agree (5)" or "Somewhat agree (4)" for the item regarding redistribution of wealth, and respondents were categorized into Class 1 or 2 based on the posterior probabilities of each latent class. According to Table 3, the number of supporters of redistribution of wealth forming Class 1 is 1772, and the number of supporters of such policies in Class 2 is 1135. While the latter group is smaller than the former, the members of Class 2 account for a substantial portion of the supporters of redistribution of wealth. Therefore, the influences affecting Class 2 cannot be overlooked when considering the mechanism that generates support for redistribution of wealth.

Except for the rates of women, the self-employed, and non-regular employees, there are no significant differences in demographic characteristics or socio-economic status between the supporters of redistribution of wealth in Classes 1 and 2. On the other hand, in terms of key independent variables (support for free-market competition and reliance on leaders/experts), there are prominent differences between them.

First, Table 3 clarifies that the supporters of redistribution of wealth in Class 1 (supporters of redistribution of wealth based on ideology) have a low average of support for free-market competition compared to those in Class 2 (supporters of redistribution of wealth based on self-interest). The mean of support for redistribution of wealth among those in Class 1 is 2.33, whereas it is 4.22 among those in Class 2. This difference is clearly statistically significant. Considering the relationship between support for redistribution of wealth and support for free-market competition, it can be said that, while the supporters of redistribution of wealth in Class 2 are less likely to have consistent preferences regarding social policies, those in Class 1 are more likely to have consistent ones.

Next, for the average of reliance on leaders/experts, the value among supporters of redistribution of wealth in Class 2 is higher than that of Class 1. This suggests that the members of Class 2 are more likely to be influenced by opinions of leaders/experts than the members of Class 1. Therefore, if there are disagreements among leaders and experts in various fields, the members of Class 2, who tend to determine their preferences regarding social policies based on self-interest, will have relatively contradictory or inconsistent preferences regarding them. In the modern world, it is difficult for ordinary people to correctly assess what is truly in their best interest. Thus, by deferring to the opinions of various leaders/experts simultaneously, they might unintentionally develop inconsistent preferences regarding social policies.

Unfortunately, these results are the opposite of the expected results from Hypothesis 2 and 3 of this study; however, they present many implications for the formation of preferences regarding social policies.

**Table 3.** Descriptive statistics of supporters of redistribution (Class 1 and Class 2).

| | Class 1 (N = 1772) | | | | Class 2 (N = 1135) | | | | Welch's *t*-Test (df) |
|---|---|---|---|---|---|---|---|---|---|
| | **Mean/Ratio** | **SD** | **Min** | **Max** | **Mean/Ratio** | **SD** | **Min** | **Max** | |
| Support for free competition | 2.33 | 0.83 | 1 | 4 | 4.22 | 0.55 | 1 | 5 | −73.52 *** (2901) |
| Respect for old ways | 2.26 | 1.05 | 1 | 5 | 2.29 | 1.12 | 1 | 5 | −0.69 (2300) |
| Reliance on experts/leaders | 2.54 | 1.12 | 1 | 5 | 2.74 | 1.18 | 1 | 5 | −4.45 *** (2331) |
| Age | 54.93 | 15.8 | 20 | 80 | 54.31 | 17.12 | 20 | 80 | 0.97 (2275) |
| Women | 0.60 | | | | 0.49 | | | | 6.09 *** (2378) |
| Primary education | 0.17 | | | | 0.16 | | | | 0.59 (2453) |
| Middle education | 0.64 | | | | 0.63 | | | | 0.54 (2406) |
| Higher education | 0.19 | | | | 0.21 | | | | −1.18 (2357) |
| Married | 0.72 | | | | 0.71 | | | | 0.49 (2402) |
| Unmarried | 0.15 | | | | 0.16 | | | | −0.74 (2370) |
| Divorced/bereaved | 0.13 | | | | 0.13 | | | | 0.15 (2428) |
| Self employed | 0.12 | | | | 0.17 | | | | 3.61 *** (21678) |
| Regular employee | 0.27 | | | | 0.27 | | | | 0.07 (2419) |
| Non-regular employee | 0.25 | | | | 0.17 | | | | 4.87 *** (2638) |
| No job | 0.33 | | | | 0.36 | | | | −1.56 (2383) |
| Job seek | 0.03 | | | | 0.03 | | | | 0.34 (2477) |
| Upper white collar | 0.12 | | | | 0.13 | | | | −1.17 (2327) |
| Lower white collar | 0.21 | | | | 0.18 | | | | 2.31 * (2532) |
| Blue collar | 0.30 | | | | 0.30 | | | | 0.35 (2427) |
| Household income (log) | 6.04 | 0.66 | 0.00 | 8.52 | 6.01 | 0.74 | 0 | 8.78 | 0.91 (2234) |

* $p < 0.05$, *** $p < 0.001$.

*4.4. Summary*

By using finite mixtures of regression models, two latent classes were extracted from the SSM 2015: One that determines its preferences regarding social policies based on self-interest, and another that does so based on ideology. This result seems to support Hypothesis 1 of this study. Regarding Hypothesis 2, however, someone who supports redistribution of wealth based on self-interest is less likely to be a consistent supporter of such policies, as he/she is more likely to support free-market competition and redistribution of wealth simultaneously. Regarding Hypothesis 3, someone who supports redistribution of wealth based on ideology is more likely to be a consistent supporter of such policies. If a member of the group that determines its preferences regarding social policies based on ideology supports free-market competition, she/he will be less likely to support redistribution of wealth as well.

Thus, Hypotheses 2 and 3 must be corrected as follows. In the analyses of this study, respect for old ways and reliance on leaders/experts were viewed as variables related to authoritarian attitudes. Moreover, it was assumed that individuals inclined to express ideological attitudes tended to form preferences regarding social policies based on the opinions of authorities instead of self-interest. However, respect for old ways and reliance on the opinions of leaders/experts have different implications. In order to correctly understand the formation of preferences regrading social policies, a distinction should be drawn between respect for old ways and reliance on the opinions of leaders/experts. It can be inferred that individuals who respect old ways form their preferences regrading social policies based on a clear principle. Therefore, their preferences tend to be consistent. On the other hand, it can be inferred that individuals who rely on the opinions of leaders/experts when forming their social policy preferences do so opportunistically. Therefore, their preferences in this area might lack consistency.

If individuals can easily judge which policies are best-suited to furthering their interests, they will express consistency in their preferences regarding social policies. In the contemporary world, it has become difficult for individuals to correctly discern the impact of social policies on their everyday lives. In order to assess the influence or impact of social policies, therefore, they must rely on discourses among political leaders or intellectuals. If there is no clear consensus among political leaders, religious leaders, or intellectuals about the impact or implications of social policies, the discrepancies in their opinions will generate contradictory beliefs related to social policies among individuals who make decisions based on self-interest.

## 5. Discussion

To clarify the latent structure of support for redistribution of wealth, data from the SSM 2015 were analyzed using finite mixtures of regression models. As a result, two latent groups were extracted successfully. One reflects a social mechanism that generates preferences regarding social policies based on self-interest and the other reflects one that does so based on ideology. Certainly, self-interest has significant implications for the formation process of preferences regarding social policies. However, "self-interest" does not hold the same meaning for all individuals. When individuals determine their preferences regarding social policies, some tend to consider self-interest carefully, while others tend not to. Rather, the latter group tends to care more about whether a social policy is suited to its beliefs (i.e., in the old ways).

However, those who determine their preferences regarding social policies based on ideology are more likely to have consistent preferences in this area. If they support free-market competition, they are less likely to support redistribution of wealth. If they do not support free-market competition, they are more likely to support redistribution of wealth. Therefore, ideological attitudes are not a cause of simultaneous support for redistribution of wealth and free-market competition. Rather, those who decide their preferences regarding social policies based on self-interest are less likely to have consistent preferences in this area. They tend to support redistribution of wealth and free-market competition simultaneously. This result was unexpected based on the theory and hypotheses of this study.

Here, it should be noted that the members of the group that determines its preferences regarding social policies based on self-interest are relatively likely to rely on the opinions of leaders/experts compared to the members of the group that expresses ideological attitudes towards ideology. In other words, by partially deferring to the opinions of leaders/experts, they judge which policies are suited to their interests. At first, it was implicitly assumed that rational individuals knew which policies were suited to furthering their interests. However, this was not always true. In order to estimate the potential impact of social policies on their lives, individuals might rely on the opinions of leaders/experts sometimes [55,56]. Thus, if there was a clear consensus among leaders and experts about the impact of social policies on people's lives, individuals who decided their social policy preferences based on self-interest would have consistent preferences in this area. However, if there was not a clear consensus among leaders, such individuals might have seemingly confusing or irrational preferences regarding social policies.

Even though people are rational individuals, they may have confusing or irrational preferences when the information that they have received about the influence of social policies includes contradictions caused by discrepancies among political leaders, religious leaders, or intellectuals. Therefore, an inconsistent preference, such as simultaneous support for redistribution of wealth and free-market competition, should be interpreted as a sign of a lack of consensus regarding social policies. Nobody can accurately assess what the merits/demerits of redistribution of wealth or the merits/demerits of free-market competition are.

On the other hand, individuals who determine their preferences regarding social policies based on ideology tend to have consistent preferences regarding them. However, their preferences are not the result of careful examination based on adequate information about the influence of each social policy. Such individuals simply obey a clear principle (e.g., the old ways), and they cannot react to various social changes in the contemporary world immediately. It is thought as a kind of cognitive misers [57]. Therefore, it can be said that, although their preferences regarding social policies appear to be rational from the standpoint of consistency, they might not be genuinely rational. Rather, the policy preferences of individuals who prioritize ideology as well as those of individuals who prioritize self-interest may lack a robustly rational foundation.

The theory and hypotheses of this study were constructed without considering the difference between authoritarian tendencies regarding social norms and those regarding attitudes towards leaders/experts. As a result, the theory and hypotheses failed to explain inconsistent preferences regarding social policies, such as simultaneous support for redistribution of wealth and free-market competition. In order to correctly analyze the formation process of preferences regarding social policies, a distinction must be drawn between authoritarian tendencies regarding social norms and those regarding attitudes toward leaders/experts. Although individuals may want to determine their preferences regarding social policies based on a rational assessment of their interests, if they do not have adequate information about the impact of each social policy, they may have seemingly irrational preferences in this area [58,59]. Rather, self-interested individuals are more likely to have inconsistent preferences regarding social policies when the impact of such policies on ordinary people is uncertain.

*Limitations*

Lastly, the limitations of this study must be highlighted. In this study, a respondent's support for social policies (redistribution of wealth and free-market competition) was measured according to only two indicators. In this study, respondents' support for social policies (the redistribution of wealth and free-market competition) was measured according to two indicators only. However, these indicators might not demonstrate the correct redistribution of wealth and free-market competition. Yet, as the SSM 2015 focused on social stratification and mobility, it did not contain adequate indicators related to preferences regarding social policies other than these. As the structure of individual preferences regarding social policies has several components and the formation process is complicated, it is problematic to gauge respondents' preferences regarding social policies using only these indicators.

In future research, individual preferences regarding social policies should be determined using more refined scales.

Furthermore, the data from the SSM 2015 used in this study are cross-sectional, and the survey was conducted in 2015. While the data from the SSM 2015 could be used to depict the structure of support for redistribution of wealth in the year when it was conducted, historical changes in the structure of support for such policies cannot be examined using only its data. In other words, the structure of support for redistribution of wealth presented in this study may be fragile and temporal. To enhance the robustness of the structure of support for redistribution of wealth and trace historical changes in that structure, longitudinal or panel data of individual preferences regarding social policies are needed.

In addition, this study is based on data from only Japan. Therefore, it cannot be said with certainty whether the findings of this study could be applied to other developed countries. In fact, previous studies argue that support for redistribution of wealth may be affected by various social contexts [24,28,30] and economic conditions [3,8,9]. Occasionally, such effects may operate on people selectively, depending on their social status [17]. Therefore, it is possible that the structure of support for redistribution of wealth found in Japanese society might be not be universal. For example, the structure of support for redistribution of wealth presented in this study might only be observed in East Asian countries, which can be characterized as having authoritarian regimes. To confirm the generality of the explanation offered in this study, the structure of support for redistribution of wealth in other countries must be examined using various datasets.

Although this study suffers from a few inevitable limitations regarding the representativeness of its findings, it succeeds in presenting the heterogeneity of supporters of redistribution of wealth in Japan. The findings of this study are also significant in explaining the formation process of individual preferences regarding social policies. Therefore, to make use of the findings of this study, its limitations must be overcome through further research.

## 6. Conclusions

Since 2012, Shinzo Abe, the president of the Liberal Democratic Party and the prime minister of Japan, has exerted his political power in a stable manner. As a conservative leader, he has promoted market-oriented policies to stimulate economic growth, enjoying popularity among the Japanese people. On the other hand, under the situation of a stagnating economy and an aging population, social inequality in Japan has widened [60]. For example, Tachibanaki [61] maintained that income equality in Japan has been expanding since the 1980s. Kariya [62] pointed out that educational inequality in Japan has grown, and Tarohmaru [63] argued that disparities between regular and non-regular employees have been more serious with the progression of globalization. As a result, many Japanese have begun to think that issues related to social welfare are important to them. However, such opinions do not seem to be influencing the actual policies adopted by the government, and the social welfare regime in Japan continues to grapple with several problems. Undoubtedly, the country is experiencing a discrepancy between the social welfare regime and support for redistribution of wealth. This can be viewed as a typical case of a discrepancy between the social regime and the people's preferences regarding social policies.

In the context of such a discrepancy between the social welfare regime and support for redistribution of wealth, we must investigate simultaneous support for redistribution of wealth and free-market competition. According to the results of the analysis presented in this study, a significant fraction of the supporters of redistribution of wealth also supports free-market competition. It should be noted that simultaneous supporters of redistribution of wealth and free-market competition are not irrational individuals. It is thought that, while they are inclined to rationally determine their preferences regarding social policies based on self-interest, they often cannot acquire adequate information to do so. The more uncertain and obscure each social policy's impact is, the more frequently such inconsistent preferences appear in society.

Therefore, as supporters of redistribution of wealth based on ideology determine their social policy preferences according to a clear principle, they tend to be consistent. However, as such individuals' preferences are determined in accordance with old traditions automatically and mechanically, their seemingly consistent preferences do not necessarily have rational grounds. Rather, individuals must have access to accurate, adequate information about the influence and impact of social policies on their lives. Based on such information, the public can finally develop fully consistent, rational preferences regarding social policies.

**Funding:** This research was funded by JSPS KAKENHI, JP18H03647 and JP19H00609.

**Acknowledgments:** This research is supported by JSPS KAKENHI (JP18H03647 and JP19H00609), and I thank the 2015 SSM Survey Management Committee for allowing me to use the SSM 2015 data (Ver. 070). And, I would like to thank anonymous reviewers of *Societies* for their helpful and constructive comments.

**Conflicts of Interest:** The author declares no conflict of interest.

## Appendix A

**Table A1.** Results for the Model without the Income Variable.

| | Model without Income Variable | | | | | |
|---|---|---|---|---|---|---|
| | Class 1 | | | Class 2 | | |
| **Variable** | **Coeff.** | | **SE** | **Coeff.** | | **SE** |
| (Intercept) | 5.127 | *** | 0.136 | 1.647 | *** | 0.225 |
| Age | 0.002 | | 0.002 | 0.005 | * | 0.002 |
| Women | 0.155 | *** | 0.039 | 0.048 | | 0.065 |
| (Married) | | | | | | |
| Unmarried | −0.012 | | 0.053 | 0.120 | | 0.095 |
| Divorced/bereaved | 0.044 | | 0.060 | 0.166 | | 0.092 |
| Primary education | 0.144 | * | 0.065 | 0.078 | | 0.091 |
| (Middle education) | | | | | | |
| Higher education | −0.026 | | 0.042 | −0.282 | *** | 0.077 |
| (Regular employee) | | | | | | |
| Self employed | −0.176 | ** | 0.060 | 0.026 | | 0.101 |
| Non-regular employee | −0.062 | | 0.053 | 0.092 | | 0.091 |
| No job | −0.104 | | 0.062 | 0.254 | * | 0.105 |
| Job seeker | −0.161 | | 0.102 | 0.424 | * | 0.203 |
| Upper white | −0.114 | | 0.056 | −0.103 | | 0.097 |
| (lower white) | | | | | | |
| Blue collar | 0.038 | | 0.052 | 0.176 | | 0.090 |
| Household income [log] | | | | | | |
| Support for free competition | −0.672 | *** | 0.026 | 0.344 | *** | 0.056 |
| Respect for old ways | 0.099 | *** | 0.021 | 0.010 | | 0.034 |
| Rely on experts/leaders | 0.014 | | 0.018 | 0.069 | * | 0.030 |
| Π | | 0.608 | | | 0.392 | |
| AIC | | | | 13,869.32 | | |
| BIC | | | | 14,098.14 | | |
| N | | | | 7021 | | |

\* $p < 0.05$, \*\* $p < 0.01$, \*\*\* $p < 0.001$.

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
