# Peer review of "Two Types of Support for Redistribution of Wealth: Consistent and Inconsistent Policy Preferences"

_societies, doi:10.3390/soc10020043_

Round 1

Reviewer 1 Report

I read the manuscript “Two Types of Support for Redistribution: Consistent and Inconsistent Policy Preferences” with great interest. The data source and the idea of different ‘types’ of redistributive preferences make the basis for new and informative social research. I am not clear on the contribution of the paper to social science however. It seems to follow different goals or is not always clear on the link between theory and research design. I list some serious and some minor concerns below.

Serious Concerns

  1. The author uses latent class modeling to identify two types of mechanisms that lead to support for redistribution in the Japanese population – self-interest and ideology, or the author identifies two types of support for redistribution – one based on self-interest and the other on ideology. This difference is subtle but crucial to the theoretical conclusions of the paper. The question is whether the author is trying to explain how and why people form preferences for redistribution (a path model where x causes y theoretically), or if the author is trying to demonstrate two different types of attitudes that exist (a measurement model where y is latent and measured by several observed survey questions so that an observed set of y were caused by latent y). Both models can only exist simultaneously if all variables measuring latent variable y remain on the ‘left-hand’ side of the equation, and all variables considered as predictors or causes of y remain on the right-hand side of the equation. The problem here is that the author used the variables ‘support for redistribution’ and ‘support for free competition’ at least together to identify latent classes (both y variables), but then used ‘support for free competition as a predictor variable (an x variable). If I did not understand this correctly, I would ask that the author does more work explaining his/her methods. If I understood correctly there is endogeneity bias. Either the variables are part of the measurement model identifying the latent classes, or they are part of the path model predicting the dependent variable latent classes. When the independent variables are also part of the dependent variable’s measurement then of course you get large and significant effects, but these cannot be represented as theoretical mechanisms, they are statistical artifacts due to endogeneity.
  2. The other serious concern has to do with the survey question wording. I realize that the two dependent variable questions are translations, but I am not 100% sure the author interpreted them correctly. The first question “Rather than protecting free-market competition, it is more important to eliminate differences among people” does not ask if people are for or against free-market competition, instead it asks if reducing group differences is more important than free-market competition. The second question “To the extent that opportunities are equally available, we must accept the disparity in wealth that results from competition” also does not ask people to be for or against free-market competition, it asks if we must accept unequal outcomes even when opportunities are equally available. If I understand correctly, this is a question on meritocracy. That some people have greater abilities than others and will end up with greater rewards in the market (even if all have the same chances). The author has interpreted the first question as support for redistribution and the second as support for free-market competition. I am not convinced this is exactly correct.

Minor Concerns

  1. The author often refers to redistribution of “wealth”, but the question ‘support for redistribution’ (if translated correctly) only asks about differences between social groups. It does not mention “wealth”. It also does not mention “income”. How can we be sure this is measuring support for redistribution? And if it is, is it wealth or income? These are very different things.
  2. The author imputes a great deal of missing values for income. My understanding is that multiple imputation or a full-information maximum likelihood estimator is best for this task. The author has instead opted for single imputation which has a great risk of introducing bias into the regressions. I would like to see results of an analysis in an appendix without the income variable (so all possible cases) AND on a model without the imputation (the reduced number of cases).
  3. Authority versus ideology. The author switches between “ideology” and “authoritarianism” in the paper. I think of ideology as either political (like left-right, or liberal-conservative-nationalist for example) and preferences for authority as something else. Maybe just using one term would make things clearer?

Author Response

Reply to Reviewers

I appreciate the valuable comments from the reviewers. On the basis of their remarks, I have revised my manuscript as follows:

Reviewer 1:

  1. The question is whether the author is trying to explain how and why people form preferences for redistribution (a path model where x causes y theoretically), or if the author is trying to demonstrate two different types of attitudes that exist. … The problem here is that the author used the variables ‘support for redistribution’ and ‘support for free competition’ at least together to identify latent classes (both y variables), but then used ‘support for free competition as a predictor variable (an x variable).

   Corresponding to Reviewer 1’s comments, I have added the following to Subsection 4.2:

Then, it needs to be noted that models in Table 2 include “support for free-market competition” as an independent variable. If it is assumed that these models predict support for social policies, this means that the variance in endorsements of social policies is partially explained by their support. Considering the possibility of endogeneity bias, therefore, the highly significant effects of support for free-market competition on support for the redistribution of wealth should not be overestimated Nevertheless, discrepancies in the directions of the effects between Classes 1 and 2 should be paid more attention to. The direction of the effect of support for free-market competition on support for the redistribution of wealth in Class 1 is opposite to that of the effect in Class 2. This implies that Classes 1 and 2 have different internal structures for support for social policies.

  1. The author has interpreted the first question as support for redistribution and the second as support for free-market competition. I am not convinced this is exactly correct.

   In line with Reviewer 1’s comments, I have added the following to the Limitations section:

In this study, respondents’ support for social policies (the redistribution of wealth and free-market competition) was measured according to two indicators only. However, these indicators might not demonstrate the correct redistribution of wealth and free-market competition. Yet, as the SSM 2015 focused on social stratification and mobility, it did not contain adequate indicators related to preferences regarding social policies other than these. As the structure of individual preferences regarding social policies has several components and the formation process is complicated, it is problematic to gauge respondents’ preferences regarding social policies using only these indicators. In future research, individual preferences regarding social policies should be determined using more refined scales.

  1. The author often refers to redistribution of “wealth”, but the question ‘support for redistribution’ (if translated correctly) only asks about differences between social groups.

   According to Reviewer 1’s comments, I have added the following to the manuscript:

Since the respondents were asked about this item just after they were asked about the item related to “support for free-market competition,” it is naturally assumed that, even though this item does not refer to “wealth,” the respondents perceived it as centering on differences in wealth among people.

  1. I would like to see results of an analysis in an appendix without the income variable (so all possible cases) AND on a model without the imputation (the reduced number of cases).

I present the results of the models without imputation and without the income variables in Appendix A.

  1. The author switches between “ideology” and “authoritarianism” in the paper. Maybe just using one term would make things clearer?

 As much as possible, I have modified the entire manuscript by using one term (ideology) only.

I hope that the revised manuscript is acceptable for publication and appreciate your reconsideration. Thank you very much.

Reviewer 2 Report

In the manuscript entitled "Two Types of Support for Redistribution: Consistent and Inconsistent Policy Preferences" the dynamics of support for redistribution of wealth is considered. The authors formulated 3 hypotheses and tried to confirm or disprove it by analyzing data from the National Survey of Social Stratification and Social Mobility in 2015 (SSM 2015).

In my opinion, this is a carefully prepared text.

My general comments are as follows:

  1. In the introduction, the structure of the preference of redistribution and attitude to the free-market competition is analyzed but there is no justification of the research. Why the results are important and who can use them in the future.
  2. The Theory and hypothesis section is presented.
  3. The main drawback of Discussion is the very small number of references to the results of other studies.

My detailed comments are as follows:

Line 2: The title should contain information about the kind of redistribution, ie "wealth" as I understand.

Line 20: According to Instruction for Authors the aim should be placed at the end of the section.

Line 180 and those which concern the survey's statements: Highlighting survey statements from the remaining text will facilitate reading and understanding of information.

Line 328: Please explain, what you mean by “incorrect interest”.

Lines 513-5015: Please give an example (and reference) of "social inequalities" and "issues related to social welfare".

Author Response

Reply to Reviewers

I appreciate the valuable comments from the reviewers. On the basis of their remarks, I have revised my manuscript as follows:

Reviewer 2:

  1. In the introduction, the structure of the preference of redistribution and attitude to the free-market competition is analyzed but there is no justification of the research.

In line with Reviewer 2’s comments, I have added the following paragraph to the last part of the Introduction:

Therefore, the author of this study aims to clarify the latent structure of support for the redistribution of wealth and, in doing so, specify the process of forming preferences for social policies. Hence, we clearly understand why inconsistent preferences for social policies can be generated by rational individuals, and why their preferences tend not to influence actual social policies. Moreover, on the basis of this finding, we will find a way to overcome problems deduced from seemingly irrational preferences for social policies.

  1. The main drawback of Discussion is the very small number of references to the results of other studies.

 I have added some references to the Discussion.

  1. The title should contain information about the kind of redistribution, ie "wealth" as I understand.

I have altered the title slightly according to your advice.

  1. According to Instruction for Authors the aim should be placed at the end of the section.

I have replaced the aim at the end of the Introduction.

  1. Highlighting survey statements from the remaining text will facilitate reading and understanding of information.

I have highlighted the survey statements using italics.

  1. Please explain, what you mean by “incorrect interest”.

I have changed “incorrect interest” to “inconsistent.”

  1. Please give an example (and reference) of "social inequalities" and "issues related to social welfare".

On the basis of Reviewer 2’s remarks, I have added the following to the manuscript:

On the other hand, under the situation of a stagnating economy and an aging population, social inequality in Japan has widened [60]. For example, Tachibanaki [61] maintained that income equality in Japan has been expanding since the 1980s. Kariya [62] pointed out that educational inequality in Japan has grown, and Tarohmaru [63] argued that disparities between regular and non-regular employees have been more serious with the progression of globalization.

I hope that the revised manuscript is acceptable for publication and appreciate your reconsideration. Thank you very much.

Round 2

Reviewer 1 Report

I feel that the author has made high quality changes to the manuscript and has presented the findings in a humble manner. I believe the manuscript is ready for publication. However, I have one comment that the author might consider. In the appendix the author presents the model with income imputed, and then without the income variable. I think the author wants to show a model with income imputed and without income imputed but with income in the model. If the significant effect of income only exists after imputation, this is problematic. We cannot really know the nature of missing data. We can only guess. The relationship of income in the real data should somewhat follow its relationship in the imputed data. If not, it does not change the results so much, but is an important and honest way to present the findings. In my opinion. If the author chooses not to include a model with the income variable (with lots of missing cases), I still recommend publication. 

Author Response

Reply to Reviewers

I highly appreciate the reviewers’ valuable comments. Based on the commnets, I have revised my manuscript as follows.

Responses to Reviewer 1’s Suggestion:

  1. If the author chooses not to include a model with the income variable (with lots of missing cases), I still recommend publication.

  • According to the reviewer’s suggestion, I removed the results of the model with the income variable (with lots of missing cases) from Table A1 in Appendix A.

I hope that the revised manuscript is acceptable for publication and appreciate your reconsideration. Thank you very much.